# Model of Chronic Thromboembolic Pulmonary Hypertension in Rats Caused by Repeated Intravenous Administration of Partially Biodegradable Sodium Alginate Microspheres

**DOI:** 10.3390/ijms22031149

**Published:** 2021-01-24

**Authors:** Andrei A. Karpov, Nikita A. Anikin, Aleksandra M. Mihailova, Sergey S. Smirnov, Dariya D. Vaulina, Leonid A. Shilenko, Dmitry Yu. Ivkin, Alexei Y. Bagrov, Olga M. Moiseeva, Michael M. Galagudza

**Affiliations:** 1Almazov National Medical Research Centre, 197341 St. Petersburg, Russia; freedomofstyle@yandex.ru (N.A.A.); aechika82@mail.ru (A.M.M.); saveliy.morozov.93@mail.ru (S.S.S.); uplavice@gmail.com (D.D.V.); shilenko.leo@yandex.ru (L.A.S.); moiseeva.cardio@gmail.com (O.M.M.); 2First Pavlov State Medical University of Saint Petersburg, 197022 St. Petersburg, Russia; 3N.P. Bechtereva Institute of Human Brain, Russian Academy of Science, 197376 St. Petersburg, Russia; 4Saint Petersburg State Chemical Pharmaceutical University, 197376 St. Petersburg, Russia; dmitry.ivkin@pharminnotech.com; 5Sechenov Institute of Evolutionary Physiology and Biochemistry of the Russian Academy of Sciences, 194223 St. Petersburg, Russia; aybagrov@gmail.com

**Keywords:** chronic thromboembolic pulmonary hypertension, experimental model, microspheres, pulmonary embolism, rats, sodium alginate

## Abstract

Chronic thromboembolic pulmonary hypertension (CTEPH) is a rare and life-threatening complication of pulmonary embolism. As existing animal models of CTEPH do not fully recapitulate complex disease pathophysiology, we report a new rat model for CTEPH evoked by repetitive embolization of the distal pulmonary artery branches with partially biodegradable alginate microspheres (MSs). MSs (180 ± 28 μm) were intravenously administered eight times at 4-day intervals; control animals received saline. The validity of the model was confirmed using transthoracic echocardiography, exercise testing, catheterization of the right ventricle, and histological examination of the lung and heart. The animals in the CTEPH group demonstrated a stable increase in right ventricular systolic pressure (RVSP) and decreased exercise tolerance. Histopathological examination revealed advanced medial hypertrophy in the small pulmonary arteries associated with fibrosis. The diameter of the main pulmonary artery was significantly larger in the CTEPH group than in the control group. Marinobufagenin and endothelin-1 serum levels were significantly elevated in rats with CTEPH. In conclusion, repetitive administration of alginate MSs in rats resulted in CTEPH development characterized by specific lung vasculature remodeling, reduced exercise tolerance, and a persistent rise in RVSP. The developed model can be used for pre-clinical testing of promising drug candidates.

## 1. Introduction

Pulmonary embolism (PE) ranks third in prevalence among all cardiovascular diseases [1]. The risk of venous thromboembolism in patients under 45 years of age varies in the range of 1.0–1.5 per 1000 people per year; after 40 years, this almost doubles with every 10-year increase in age. Treatment for this population thus places a heavy burden on the health care system [2]. In 4–9% of patients with acute PE, the course of the disease is complicated by the development of chronic thromboembolic pulmonary hypertension (CTEPH), associated with extremely high mortality [3]. It has been demonstrated that the ten-year survival rate of patients who have undergone PE with an average pressure in the pulmonary artery (31–40 mm Hg) is 50%; at levels between 41–50 mm Hg and >50 mm Hg, the rates are 20% and 5%, respectively [4]. Despite the high efficiency of pulmonary artery thromboendarterectomy in proximal CTEPH, most modern therapeutic approaches require improvement [5]. At the same time, pre-clinical studies of new pharmacological substances are difficult due to the lack of experimental models that sufficiently reproduce the pathophysiological mechanisms of CTEPH development, as well as the structural changes in the pulmonary vascular bed arising from CTEPH.

Currently, to model PE and CTEPH in rodents, native thrombi administered in combination with inhibitors of the fibrinolysis system, such as tranexamic acid [6,7] and polystyrene microspheres [8,9,10], are often used as embolic particles. However, the use of models with native thrombi is hampered by technological difficulties; namely, the need to form autologous thromboemboli for each animal, and the instability of pressure increases in the pulmonary artery [6,11]. The introduction of polystyrene microspheres does not fully reflect the pathogenesis of CTEPH due to the absolute insolubility of emboli. Therefore, we were interested to develop a model of CTEPH caused by repetitive intravenous administration of partially degradable emboli. For this purpose, we thought to use microspheres composed of sodium alginate. Alginates are linear biological polymers composed of 1,4-linked β-d-mannuronic acid (M) and 1,4 α-l-guluronic acid (G) residues arranged in homogenous (poly-G, poly-M) or heterogenous (MG) block-like patterns [12,13,14]. Owing to their unique capability of sol–gel phase transition, alginates can easily form various semisolid or solid structures. Further, alginates undergo in situ gelation that makes them promising materials for a wide range of applications, including injectable biopolymers for tissue engineering or targeted drug delivery systems [15,16]. Sodium alginate is one of the most extensively studied types of alginate both in the pharmaceutical and biomedical field [17]. Additionally, sodium alginate is regarded as biocompatible, nonimmunogenic, nontoxic, and not degradable in mammalian gastro-intestinal tract material [18].

Therefore, the aim of this study was the development and validation of a rat CTEPH model based on the recurrent embolization of pulmonary artery branches with partially biodegradable sodium alginate microspheres (MSs).

## 2. Results

### 2.1. Animal Survival

During modeling, the survival rate in the CTEPH group was 56% (Appendix A). The causes of death of animals in all cases were ischemic stroke and acute heart failure. The rats died within 2–3 min from the moment of MS injection. In the control group, there was no loss of animals during the experiment.

### 2.2. Treadmill Test

According to the treadmill test, the distance covered at all observation points in the experimental group was significantly lower than in the control group (*p* < 0.01 for all observation points) (Figure 1A). In addition, the dynamics of the gain of exercise tolerance in the experimental group were noted from weeks 0–4 after microsphere administration (*p* < 0.01). From the 5th week, there was a reverse trend towards a decrease in the distance covered; statistical analysis revealed significant differences between 4 and 18 weeks (*p* = 0.01), and between 18 and 5 weeks (*p* = 0.04). There were no significant differences between the observation points in the control group.

### 2.3. Transthoracic Echocardiography (TTE)

According to TTE, there was a statistically significant increase in the diameter of the main pulmonary artery (MPA) in the experimental group between 4 (*p* = 0.031), 5 (*p* = 0.004), and 6 (*p* = 0.005) weeks of observation when compared with the 0-week group after the last injection of microspheres. In addition, there was a significant increase in the MPA diameter in the experimental groups compared with the control group after 5 (*p* = 0.041) and 6 (*p* = 0.02) weeks of observation (Figure 1B).

In the experimental group, compared with 0 weeks, there was a tendency (*p* = 0.062) toward decreasing right ventricle (RV) systolic function assessed by systolic excursion of the tricuspid valve annulus (TAPSE) by week 6 of observation. At the same time, there was a significant difference between the TAPSE values in the control and experimental groups 6 weeks after the last injection of microspheres (*p* = 0.025) (Figure 1C).

Left ventricle (LV) systolic function, measured using shortening fraction (FS) at all points of observation, remained within the normal range in both the experimental and control groups (Figure 1D).

The values of peak flow rate in the pulmonary artery and RV outflow tract, size of the RV outflow tract, and heart rate did not differ between the experimental and control groups.

### 2.4. Cardiac Catheterization with Manometry

RV systolic pressure (RVSP) according to cardiac catheterization data immediately after the last injection of microspheres (week 0) and after 18 weeks was significantly higher in the experimental group than in the control group (*p* = 0.003 and *p* = 0.004, respectively) (Figure 2A). Similar data were obtained for the RVSP/cardiac output (CO) ratio; significant differences were obtained for the 0- and 18-week observation points (*p* = 0.003 and *p* = 0.004, respectively) (Figure 2B).

In the experimental group, a significant increase in RV mean pressure (RVMP) was noted at 0 (*p* = 0.001), 3 (*p* = 0.005), 5 (*p* = 0.05), and 18 (*p* = 0.003) weeks of observation, compared with the control group (*p* = 0.042 and *p* = 0.048, respectively).

There were no significant differences in the parameters of left atrial pressure (LAP) and mean blood pressure (Table 1).

### 2.5. Histological Examination of the Lung Vessels 

During histological examination, 1999 vessels were analyzed. The percentage of vessels in which a microsphere was immediately detected in the lumen after the last embolization was 36.4 ± 1.4%. Considering the data from the subsequent observation points, a graph of microsphere biodegradation was constructed (Figure 3B).

The percentage of vessels with signs of leukocytic infiltration was: 77.7 ± 4.7%: 0 weeks, 62.8 ± 10.9%: 1 week, 20.2 ± 4.8%: 2 weeks, 19.6 ± 15.4%: 3 weeks, 0.6 ± 1.3%: 4 weeks, 1.7 ± 2.4%: 5 weeks, no leukocyte infiltration was detected at 6 weeks, and 5.1 ± 12.6%: 18 weeks.

Taking into account the significant severity of inflammatory changes in the vascular wall at the 0- and 1-week observation points, the severity of hypertrophic changes in the vessels without leukocyte infiltration was measured, starting from the 2nd week of observation. The hypertrophy index of the vascular wall of the pulmonary artery branches at all observation points and in all subgroups in the experimental group was significantly higher than in the control group (*p* < 0.001). Simultaneously, there was a significant increase in indicators by 5 and 6 weeks, compared with 2 weeks (*p* < 0.05). By the 18th week of observation, the vascular wall hypertrophy index decreased in the subgroup of vessels with an outer diameter of 80–119 μm when compared to weeks 5 and 6 (*p* = 0.02, *p* = 0.04, respectively), as well as in the subgroup of vessels with an outer diameter of 40–79 μm when compared to week 6 (*p* = 0.004).

The severity of fibrotic changes in the vascular wall, assessed using Heidenhain’s Azan stain, increased in parallel with the observational period (Figure 4). The percentages of collagen fibers in the vascular wall of the pulmonary artery branches were 23.8 ± 5.9%, 44.4 ± 10.7%, and 52.3 ± 5.2% at 2, 6, and 18 weeks of observation, respectively. There was a significant difference between 2 and 6 weeks (*p* = 0.005), and between 2 and 18 weeks (*p* = 0.003).

### 2.6. Histological Heart Examination 

According to the data of the heart histological examination, a significant difference in the ratio of the areas of the RV cavity to LV cavity at 6 and 18 weeks of observation was demonstrated in the experimental group compared with the control group (*p* = 0.006 and *p* = 0.027, respectively) (Figure 5).

### 2.7. Hematological and Biochemical Parameters

According to the enzyme immunoassay (ELISA), the level of endothelin-1 in the blood plasma of the experimental animals had significantly increased at 2 and 6 weeks after the last embolization (*p* = 0.008 and *p* = 0.049, respectively) (Figure 6A). Compared with the control group, a significant increase was noted in the level of C-reactive protein (CRP) at 1, 2, 4, 5, and 6 weeks of observation (*p* < 0.05) (Figure 6C). In addition, at the 4- and 5-week stages, the CRP level was significantly higher than immediately after the last microsphere administration (week 0) (*p* = 0.014, *p* = 0.038, respectively). When analyzing the level of marinobufagenin (MBG), the following values were obtained: 0.41 ± 0.05, 0.48 ± 0.01, and 0.31 ± 0.01 for the control, 3-week, and 18-week groups, respectively. Thus, a significant increase in the level of MBG was observed during the period of CTEPH development (*p* < 0.05). At follow-up (18 weeks), there was an inversed trend towards a decrease in MBG levels compared with the control group (*p* < 0.05). A similar trend was observed when determining the level of blood leukocytes (Figure 6E); a significant increase was noted at 1, 3, 4, 5, and 6 weeks of observation, compared with the control group (Appendix A). In both analyses, indicators returned to normal by 18 weeks of observation. The level of fibroblast growth factor-1 (FGF-1) significantly varied between animals, and statistical differences were not detected at any observation point (Figure 6B).

## 3. Discussion

As a result of this work, a new model of CTEPH was created, characterized by: a stable increase in pressure in the pulmonary artery; decreased exercise tolerance; endothelial dysfunction; and the appearance of histological changes in characteristics of CTEPH: a reduction of the vascular bed due to the closure of segmental, subsegmental, and distal branches of the pulmonary artery, as well as hypertrophy and fibrosis of the vascular wall, and dilatation of the right ventricle.

The leading mechanism for the formation of pulmonary hypertension in most previously developed models of CTEPH using artificial microspheres involves reducing the vascular bed of the pulmonary circulation (PC) [19,20]. However, the current understanding of CTEPH pathogenesis goes far beyond chronic obstruction caused by unfragmented thrombotic masses. A clear confirmation of this is the absence of a significant increase in pressure in the pulmonary artery of patients who had undergone a pulmonectomy [21], i.e., the absence of a clear correlation between the increase in pressure and the volume of obstruction in CTEPH. A number of studies have highlighted aseptic inflammation as a key link in pathogenesis for the development and progression of CTEPH. 

Activated immune cells, which include macrophages, T and B lymphocytes, dendritic cells, and mast cells, invade the perivascular region and release numerous cytokines and chemokines, which in turn lead to fibrosis, vasoconstriction, and endothelial dysfunction [22]. It was thus shown that CRP [23], interleukin (IL)-10 [23], monocytic chemotactic factor-1 (MCP-1) [24], macrophage inflammatory protein 1α (MIP-1α) [23], matrix metalloproteinase 9 (MMP-9) [23], CXCL-13 [25], neopterin [26], osteopontin [27], endothelial–platelet adhesion molecule 1 (PECAM-1) [28], CXCL-10 [29], CXCL-9 [29], and CCL-5 [29] are significantly increased in patients with CTEPH. Conflicting data were obtained for tumour necrosis factor α (TNF-α) [24,30,31], IL-1β [23,24], IL-6 [23,29,31], and IL-8 [29,31].

In the present model, the microspheres were designed such that they had a slow and controlled biodegradation profile; by the end of the observational period, the percentage of vascular bed obstruction of the PC could not independently determine pulmonary hypertension, highlighting other mechanisms of its formation.

Due to its relative biological inertness and ability to biodegrade (depolymerization), alginate is often used to encapsulate transplanted cells [32] and drugs [33]. At the same time, biocompatibility or/and immunogenicity of alginate is demonstrated to vary with factors like the mannuronic acid/guluronic acid (M/G) ratio and impurities. Capsules composed of G-rich alginates in combination with polylysine were proved to induce a severe inflammatory response [34], but in general the G-rich alginates seem to possess a higher biocompatibility than the M-rich polymers [35]. It is, however, apparent that other factors like the shape and size of the beads, and smoothness, composition, and viscosity of the membrane can influence alginate immunogenicity [36]. In addition, it should be noted that the mechanical properties of the alginate differ depending on the M/G ratio. Alginates with a low M/G ratio and a high proportion of guluronic blocks have been shown to form a reliable and rigid gel. In contrast, alginates with low guluronic blocks and high M/G ratios produce soft and elastic gels [37].

One of the key advantages of the microspheres is the ability to control the rate of depolymerization by adjusting the concentration of alginate and stabilizing solution. In addition, it is possible to enclose a thrombotic mass in MS with the effect of a delayed release of biologically active substances secreted by platelets (serotonin, beta-thromboglobulin, thrombospondin, calcium ions, etc.) and fibrin degradation products, which makes it possible to further increase the compliance of pathogenesis with the modeled pathology. 

According to the literature, the question of biodegradation of alginate is controversial. On the one hand, there is evidence of good biodegradation [38]; on the other hand, it is indicated that there are no enzymes that break down alginate polymer chains in mammals [39]. Ionically cross-linked alginate gels can be dissolved by release of the divalent ions cross-linking the gel into the surrounding media due to exchange reactions with monovalent cations, such as sodium ions [39]. However, our data are more likely in favor of the successful biodegradation and elimination of alginate, since we see neither its accumulation in macrophages nor granulomas.

Intravascular alginate MS, to some extent, can stimulate the activation of macrophages, fibroblasts, and myofibroblasts [40,41], however, as well as an unlysed thrombus in CTEPH. It should be noted that the number of preserved microspheres by 18 weeks was 2.4% of those available at 0 weeks of observation, thus fibrotic changes in the vascular wall cannot be explained solely by the response of the immune system and fibroblasts to the biomaterial. Consideration should be given to the role of persistent increases in pulmonary artery pressure and endothelial dysfunction.

When analyzing the data of the treadmill test (distance traveled) and cardiac catheterization (RVSP, RVSP/CO), there was a biphasic change in indicators; the initial trend towards normalization of indicators ran parallel to the biodegradation of microspheres. Starting from 4–5 weeks of observation, an opposite trend was observed regarding the biodegradation of microspheres. This indicates the emergence of other factors affecting the further progression of pulmonary hypertension, in addition to mechanical reduction of the vascular bed of the PC. A possible reason is the change in aseptic local inflammation of the vascular wall with fibrotic changes, characterizing the formation of CTEPH. In addition, persistent endothelial dysfunction, including those manifested by a stable increase in the level of endothelin-1 in the blood plasma, also contributes to vasospasm, and, indirectly, to the progression of fibrotic changes in the vascular wall. A significant role of progressive fibrotic changes in the vascular wall is demonstrated in this study by the results of a histological study using a dye on connective tissue (Heidenhain’s Azan stain).

According to the histological examination data, the vascular wall hypertrophy index systematically increased in the vessels belonging to the area of direct embolization (40–79 μm), and to a greater extent in the vessels with a large diameter. These data, along with RV dilatation, indicate the systemic nature of changes in the hemodynamics of the PC.

The controversy surrounding the decrease in the vascular wall hypertrophy index and increase in RVSP and RVSP/CO is likely associated with progressive wall fibrosis. It should be noted that fibrotic changes in the vascular wall reduce sensitivity to vasodilators of the pulmonary circulation [42], irreversibly increase pulmonary vascular resistance [43], and are critically important for the progression of CTEPH and worsening of the patient’s prognosis.

In this study, the level of MBG in blood plasma was assessed along with other factors. MBG is an endogenous cardiotonic steroid, proven to be involved in the development of fibrosis in the cardiovascular system via chronic administration of MBG in a model of renal failure in rats, as well as in preeclampsia in humans [44,45]; however, the influence and change in the level of MBG in pathologies of the PC has been little studied. Bagrov et al. [46] demonstrated a significantly lower level of digoxin-like factor in patients with complicated myocardial infarction and pulmonary edema, compared with patients with uncomplicated myocardial infarction. Kennedy et al. [47] showed that an increase in MBG in patients with heart failure is associated with a decrease in RV systolic function and a poor prognosis. In this study, an increase in the level of MBG in the 3-week subgroup was noted when compared with the control animals. At the same time, by the 18th week of observation, MBG levels returned to normal. 

The limitations of this study are the insufficient assessment of the MBG level in the early stages after embolization as well as the role of increasing this factor in the process of vascular wall fibrosis of the PC vessels. Routine coagulation parameters, D-dimer, fibrin-degradation products, and brain natriuretic peptide, unfortunately, were not measured. The study analyzed a small spectrum of cytokines. In addition, it would be appreciated to evaluate the changes in the leukocyte infiltration of the vascular wall at different points of CTEPH formation using immunohistochemistry.

This model, unfortunately, is characterized by a significant death of animals after the administration of MS, which is not entirely compatible with the requirements of 3R (Reduction, Refinement, Replacement). Nevertheless, we consider it ethically acceptable to use this model, since it is impossible to achieve the desired result in any other way, and patients with CTEPH need new effective approaches to treatment.

Thus, the developed model can be used to study the pathogenesis of CTEPH as well as to conduct pre-clinical studies of new medicinal substances aimed at both selective vasodilation of the vessels of the pulmonary circulation and improving the prognosis of survival through their antifibrotic, anti-inflammatory, and antiproliferative properties.

## 4. Materials and Methods 

### 4.1. Animals

A total of 140 male Wistar rats, with an average weight of 225 ± 28 g, were used. The animals were maintained in individually ventilated cages on a 12 h light/dark cycle and were provided with food and water ad libitum. The temperature and humidity within the cages were kept within the ranges recommended by the Guide for the Care and Use of Laboratory Animals. 

### 4.2. Embolic Particles

Partially biodegradable sodium alginate MSs were used as embolic particles in this study. MSs were obtained from ultrapure sodium alginate (M/G ratio–1.56) (Sigma-Aldrich, St. Louis, MO, USA) using a B-390 electrostatic encapsulator (Buchi, Switzerland); a 2% barium chloride solution was used as a stabilizing agent. Immediately before the administration, the MSs were washed in saline. The size of the resulting microspheres was 180 ± 28 μm. All microspheres were produced under sterile conditions.

### 4.3. Experimental Protocol

At the first stage, 100 animals were randomly selected from the study population (*n* = 120); they were injected 8 times at 4-day intervals with 9367 ± 551 MSs suspended in 1 mL of saline into the tail vein to reproduce PE. The remaining animals (*n* = 20) were injected with physiological saline according to the above protocol.

The day after the last injection of microspheres, a treadmill test was performed; equilibrium randomization of animals was performed for different points of excretion based on the results of exercise tolerance (0, 1, 2, 3, 4, 5, 6, and 18 weeks after the last injection of MSs). The animals did not receive any additional treatment, particularly with anticoagulants.

At each time point, all animals underwent a treadmill test and clinical blood test. Seven animals underwent TTE, catheterization of the heart chambers with manometry, histological examination of the lungs and heart, and assessment of the levels of endothelin-1, FGF-1, and CRP in blood plasma by enzyme immunoassay. Invasive research methods in the control group were performed 18 weeks after the last injection of saline (Figure 7).

### 4.4. Treadmill Test

A treadmill device (model LE8710, Harvard Apparatus, Holliston, MA, USA) was used to perform the exercise test. One day before the start of microsphere administration, all rats were trained under testing conditions. There was no training effect within the training. During testing, a protocol with a gradual increase in the rotation speed of the treadmill belt was used: 5 m/min increasing every 30 s, up to a speed of 40 m/min. Stimulation in the form of a direct current electrical impulse was automatically carried out with a current of 1.2 A when the animals touched the edge of the running belt. The distance that the animal ran during testing was assessed (m); the end point of the test for each animal was their inability to continue running.

### 4.5. Transthoracic Echocardiography

To perform the study, a high-resolution ultrasound unit (MyLab One Touch SL 3116, Esaote, Genoa, Italy) with a vascular linear probe (frequency: 13 MHz, scanning depth: 2 cm) was used. Before TTE, the animals were anesthetized via isoflurane inhalation using a SomnoSuite® Low-Flow Anesthesia System for gas anesthesia (Kent Scientific, Torrington, CT, USA) and placed on a heated table (TCAT-2LV controller, Physitemp Instruments Inc., Clifton, NJ, USA) in the supine position. The main parameters evaluated in the study were the: (1) diameter of the MPA (mm), (2) size of the RV outflow tract (mm), (3) peak flow rate in the pulmonary artery (m/s), (4) peak flow rate in the RV outflow tract (m/s), (5) TAPSE (mm), (6) FS (%) of the LV, and (7) heart rate (bpm).

### 4.6. Cardiac Catheterization with Manometry

Before cardiac catheterization, the animals were anesthetized by the intramuscular administration of combined anesthesia using Zoletil^®^ (Virbac, Carros, France) and Xylazine 2% (Interchemie Werken “de Adelaar” BV, Venray, The Netherlands); artificial lung ventilation was carried out through tracheal intubation using a ventilator (SAR-830/AP, CWE Inc., Ardmore, PA, USA). The following parameters of artificial lung ventilation were used: respiratory rate: 60–70/min; respiratory capacity: 3 mL/100 g of body weight. To control hemodynamic parameters of the systemic circulation, catheterization of the carotid artery and the registration of systemic arterial pressure were performed. RV manometry was performed through chest opening and RV puncture. Pressure in the left atrium was recorded using a separate puncture catheter. To register CO, a volume flow sensor (TS420 Perivascular Flow Module, Transonic, Ithaca, NY, USA) was inserted into the ascending part of the aorta, and the registration of hemodynamic parameters (RVSP, RVMP, LAP) was performed using the PhysExp Mini recorder (Cardioprotect Ltd., St. Petersburg, Russia); the RVSP/CO ratio was calculated based on the obtained data.

### 4.7. Filling the Vascular Bed of the Lungs, Followed by Histological Examination

Considering the difficulty of differentiating microvessels from the pulmonary and bronchial arteries, the vascular bed of the lungs of some animals was filled with a gelatin–acrylic mixture with a stain during histological examination, prior to organ harvesting.

The arrest of blood circulation in anesthetized rats was performed with a 10% solution of potassium chloride, injected intravenously. Immediately after cardiac arrest in the diastole phase, a mixture of 5% gelatin solution and acrylic stain (in a 5:2 ratio of two different colors), heated to 40 °C, was simultaneously injected into the carotid artery and RV under a pressure of 40–50 mm Hg; the stain was fixed to the vascular bed using 10% buffered formalin.

The right lower lobe was used for histological evaluation; this portion was divided into 4 transverse levels for analysis. Sections 3–5 μm thick were stained with hematoxylin and eosin, as well as with Heidenhain’s Azan (Biovitrum, St. Petersburg, Russia) (modification of Mallory’s method). The preparation was carried out using an Eclipse Ni-U microscope (Nikon, Tokyo, Japan) with a magnification of ×5 to ×40. The microscopic results were evaluated using the Nis Elements Br4 software (Nikon, Tokyo, Japan). In all identified vessels belonging to the branches of the pulmonary artery, the average outer diameter of the vessel and hypertrophy index, which is the ratio of the area of the vascular wall to the area of the entire vessel in percent, were determined on two distal sections of the lung. To interpret the data, all analyzed vessels were divided into groups according to their external diameter: <40 μm, 40–79 μm, 80–119 μm, and ≥120 μm; the hypertrophy index was separately determined for each subgroup.

Assessment of the severity of fibrotic changes in the vascular wall was performed using Heidenhain’s Azan stain at 2, 6, and 18 weeks after embolization. In the obtained micrographs of the vessel, a monochrome analysis of the content of blue stained fibers was carried out using ImageJ v1.50b (National Institutes of Health, Bethesda, MD, USA) (Plugin: ICH Tool Box). To assess the rate of biodegradation of microspheres in vivo at each point of excretion, the percentage of detected microspheres in two distal sections of the lung was calculated. The number of microspheres detected at week 0 immediately after the administration of the last dose of microspheres was taken as 100% (Appendix A).

To study the RV structure, the portion of the heart transversely below the left ventricular appendage was divided into three equal parts; sections 4–5 μm thick were stained with hematoxylin and eosin. The ratio of the area of the RV cavity to the area of the LV cavity was used as an estimation criterion for RV remodeling.

### 4.8. Hematological and Biochemical Parameters

Analysis of the level of leukocytes in the blood was performed using a BC-2800 Vet hemoanalyzer (Mindray, Shenzhen, China); blood for the study was taken from the tail vein. Analyses of endothelin-1, FGF-1, and CRP levels in blood plasma were performed with an automatic biochemical and ELISA analyzer (ChemWell Combi 2910, Awareness Technology, Palm City, FL, USA) using commercial kits. MBG was measured in deproteinased plasma samples by ELISA immunoassay (Shanghai BlueGene Biotech CO., LTD., Shanghai, China).

### 4.9. Statistical Analysis

Data analysis was carried out using the Statistica v7.0 (StatSoft, Tulsa, OK, USA) package.

RVSP (mmHg) 6 weeks after the last injection of MSs was selected as the primary end-point of the study. The sample size per group was determined using the following parameters: SD value (±7) on the basis of our pilot study, desired confidence level (95%), statistical power (0.9), and effect size (13) calculated according to the J. Cohen formula (CTEPH vs. healthy animals) (Power & Sample Size Program, v3.1.6). According to the results of sample size calculation, *n* = 7 per time point was found to be sufficient for making conclusions. The Kruskal–Wallis test was performed to determine differences in outcomes, followed by pairwise inter-group comparisons by a non-parametric Mann–Whitney U test; *p* values ≤ 0.05 were considered to indicate statistical significance. All data are expressed as mean ± standard deviation.

## Figures and Tables

**Figure 1 ijms-22-01149-f001:**
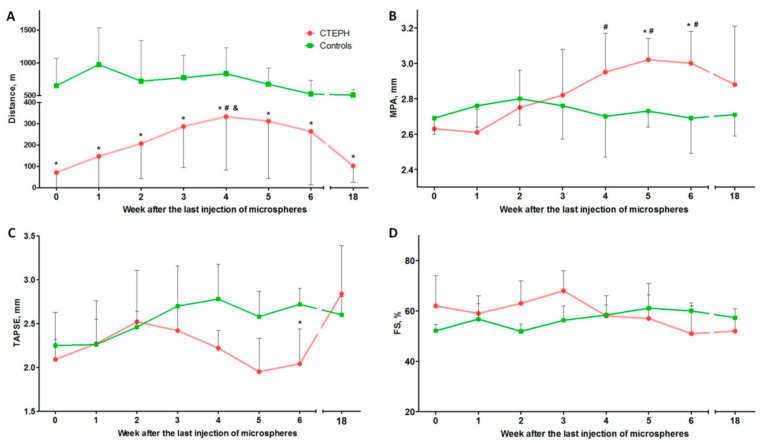
Results of non-invasive tests. (**A**) Assessment of exercise tolerance according to the treadmill test. (**B**–**D**) Results of echocardiographic examination. (**B**) The diameter of the main pulmonary artery (MPA). (**C**) Systolic excursion of the tricuspid annulus plane. (**D**) Left ventricle (LV) shortening fraction. * *p* < 0.05 in comparison with the control group. # *p* < 0.05 in comparison with the 0-week subgroup. & *p* < 0.05 in comparison with the 18-week subgroup. *n* = 7 at each time point in the chronic thromboembolic pulmonary hypertension (CTEPH) group.

**Figure 2 ijms-22-01149-f002:**
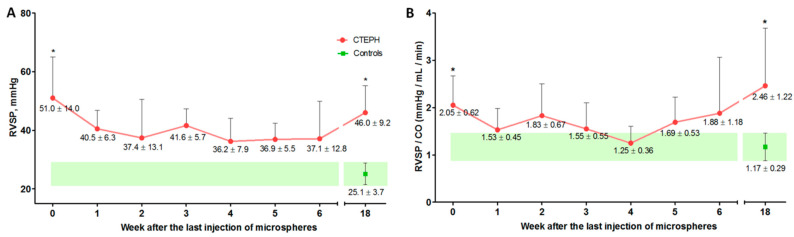
Dynamics of changes in right ventricular systolic pressure (RVSP) (**A**) and RVSP/cardiac output (CO) ratio (**B**) at different times after embolization. * *p* < 0.05 in comparison with the control group. *n* = 7 at each time point in CTEPH group.

**Figure 3 ijms-22-01149-f003:**
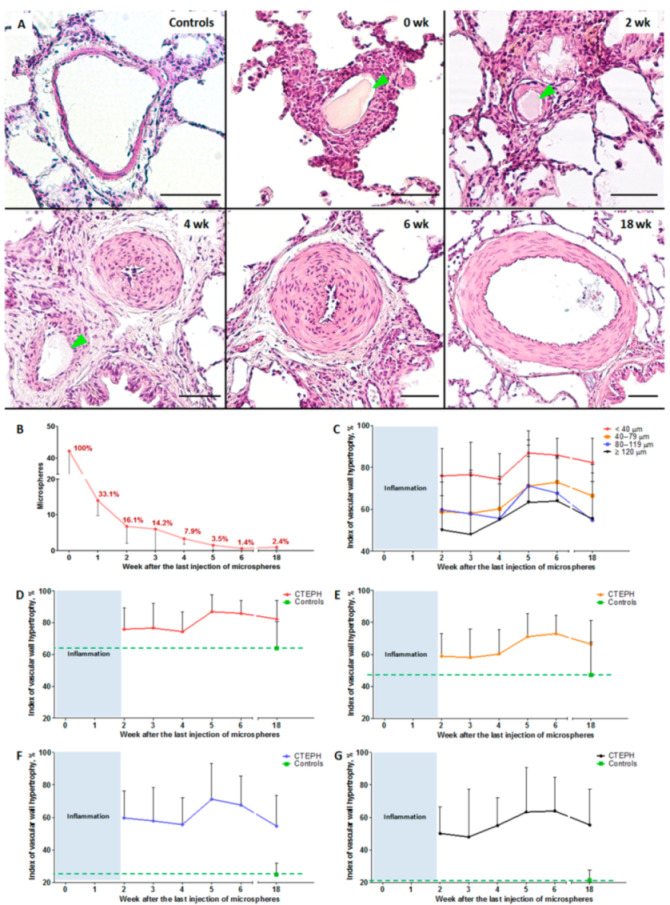
Histological examination of the lung vessels. (**A**) Representative microphotographs of the branches of the pulmonary artery at different periods of observation after embolization, staining: hematoxylin–eosin, scale bar: 50 μm, green arrows indicate microspheres in the lumen of the vessels. (**B**) Biodegradation of microspheres in the vascular bed. (**С**–**G**) Index of hypertrophy of the vascular wall of the pulmonary artery branches. (**C**) For all diameters of vessels. (**D**) For vessels with an outer diameter <40 µm. (**E**) For vessels with a diameter of 40–79 µm. (**F**) For vessels with a diameter of 80–119 µm. (**G**) For vessels with a diameter ≥120 µm. *n* = 7 at each time point in CTEPH group.

**Figure 4 ijms-22-01149-f004:**
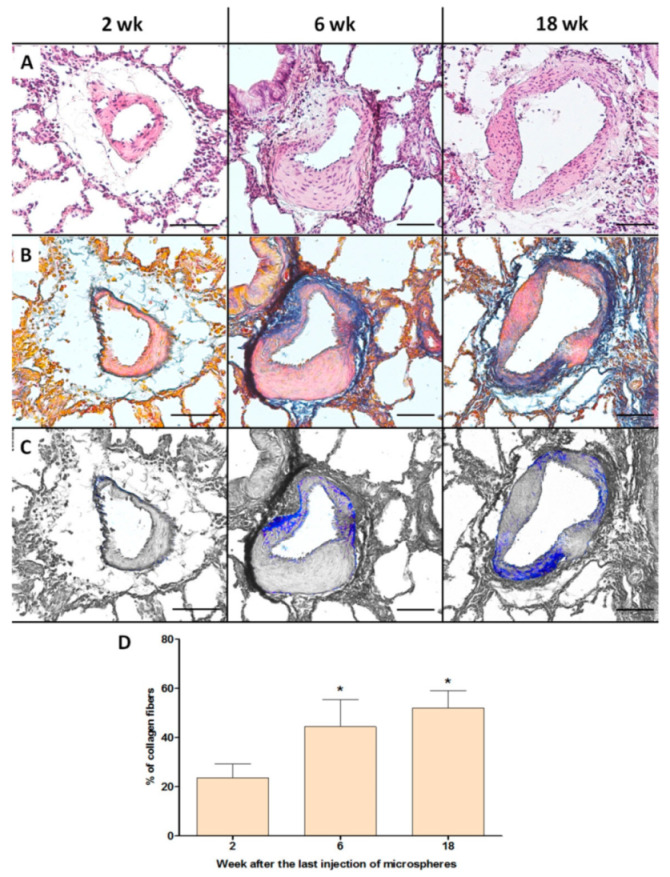
Evaluation of fibrotic changes in the vascular wall of the pulmonary artery branches at different periods of CTEPH modeling. (**A**–**C**) Representative micrographs of vessels, scale bar = 50 μm. (**A**) Haematoxylin–eosin staining. (**B**) Staining with Azan according to Heidenhain. (**C**) Monochrome isolation of collagen fibers in the structure of the vascular wall. (**D**) The percentage of collagen fibers in the vascular wall structure of the pulmonary artery branches. * *p* < 0.05 in comparison with the 2-week subgroup. *n* = 7 at each time point in CTEPH group.

**Figure 5 ijms-22-01149-f005:**
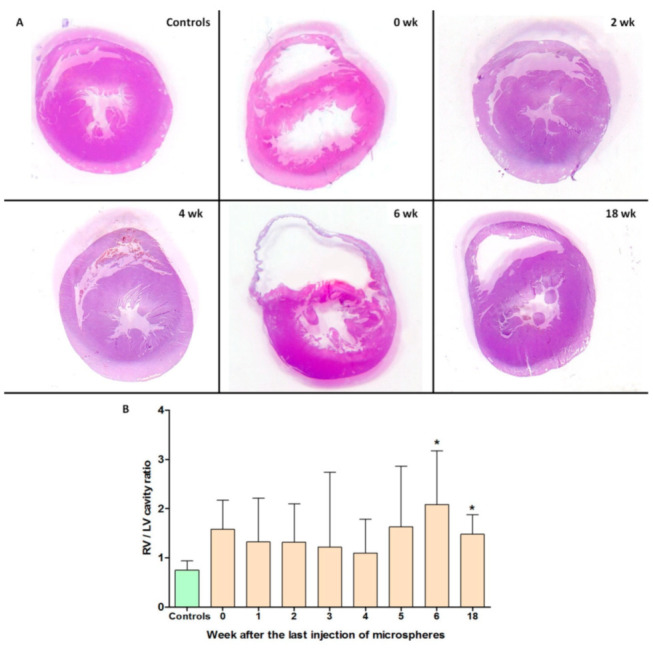
Histological heart examination. (**A**) Hematoxylin–eosin-stained representative cross-sections of the heart at different times after embolization. (**B**) The ratio of the right ventricle (RV) cavity area to the LV cavity area. * *p* < 0.05 in comparison with the control group. *n* = 7 at each time point in CTEPH group.

**Figure 6 ijms-22-01149-f006:**
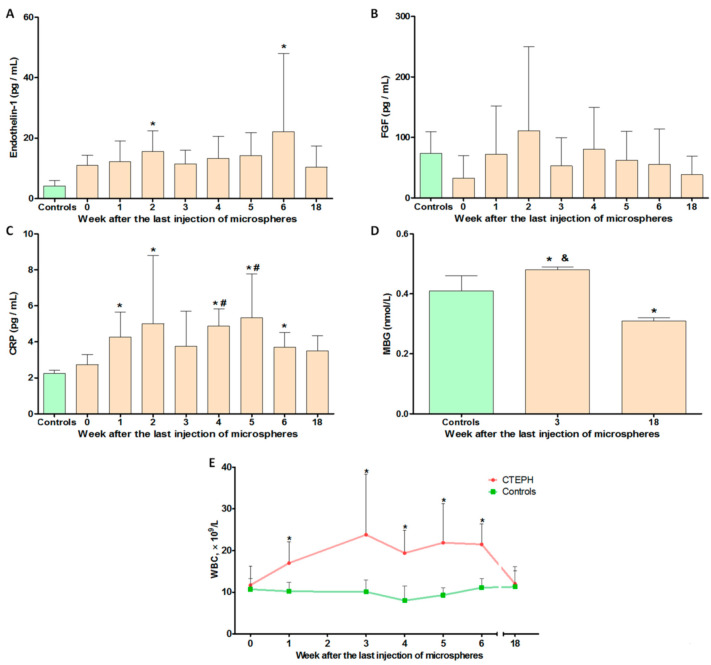
Hematological and biochemical parameters. (**A**–**D**) Study of paracrine factor content in blood plasma by ELISA. (**A**) The concentration of endothelin-1. (**B**) The concentration of fibroblast growth factor-1 (FGF-1). (**C**) The concentration of C-reactive protein (CRP). (**D**) The concentration of marinobufagenin (MBG). (**E**) The level of leukocytes in the blood. * *p* < 0.05 in comparison with the control group. # *p* < 0.05 in comparison with the 0-week subgroup. & *p* < 0.05 in comparison with the 18-week subgroup. *n* = 7 at each time point in CTEPH group.

**Figure 7 ijms-22-01149-f007:**
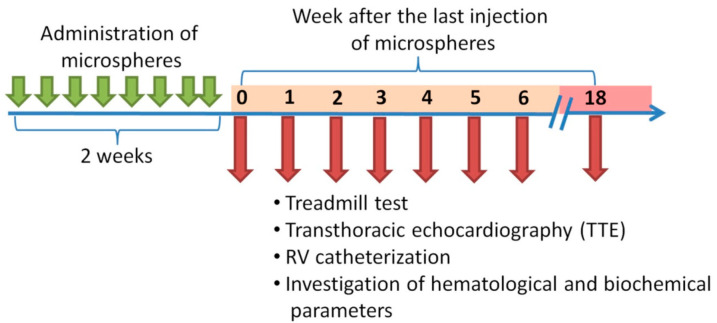
Scheme of experiment.

**Table 1 ijms-22-01149-t001:** Hemodynamic parameters in rat CTEPH model. *n = 7 at each time point in CTEPH group.*

Parameter	0 weeks	1 week	2 weeks	3 weeks	4 weeks	5 weeks	6 weeks	18 weeks	Control
RVSP (mm Hg)	51.0 ± 14.0 *	40.5 ± 6.3	37.4 ± 13.1	41.6 ± 5.7	36.2 ± 7.9	36.9 ± 5.5	37.1 ± 12.8	46.0 ± 9.2 *	25.1 ± 3.7
RVMP (mm Hg)	18.1 ± 4.4 *	14.4 ± 1.4	13.5 ± 3.4	15.6 ± 2.5 *	12.7 ± 2.1	15.3 ± 1.8 *	13.7 ± 3.9	16.1 ± 4.1 *	8.3 ± 1.8
CO (mL/min)	25.8 ± 5.5	29.5 ± 12.9	26.4 ± 9.7	29.9 ± 11.0	29.7 ± 4.2	27.7 ± 8.9	24.1 ± 10.7	24.8 ± 14.6	22.6 ± 7.2
LAP (mm Hg)	1.9 ± 0.8	2.0 ± 0.9	1.3 ± 0.6	2.3 ± 1.9	2.5 ± 1.1	1.9 ± 0.2	1.6 ± 0.7	1.7 ± 0.9	1.6 ± 0.4
RVSP/CO (mmHg/mL/min)	2.05 ± 0.62 *	1.53 ± 0.45	1.83 ± 0.67	1.55 ± 0.55	1.25 ± 0.36	1.69 ± 0.53	1.88 ± 1.18	2.29 ± 1.25 *	1.18 ± 0.37
Mean BP (mm Hg)	60.8 ± 12.2	55.5 ± 8.9	77.6 ± 13.9	60.9 ± 10.6	78.8 ± 19.1	65.7 ± 18.8	65.5 ± 26.5	58.8 ± 15.0	63.3 ± 9.2

* *p* < 0.05 in comparison with the control group. RVSP: RV systolic pressure; RVMP: mean pressure in RV; CO: cardiac output; LAP: left atrial pressure; Mean BP: mean blood pressure.

## Data Availability

The data that support the findings of this study are available from the corresponding author upon reasonable request.

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
