# Peer review of "Model of Chronic Thromboembolic Pulmonary Hypertension in Rats Caused by Repeated Intravenous Administration of Partially Biodegradable Sodium Alginate Microspheres"

_ijms, 2021, doi:10.3390/ijms22031149_

Round 1
Reviewer 1 Report
Content suggestions:
- Did the authors test also full blood count and routine coagulation parameters, D-dimers or fibrin-degradation products ? I consider them very important.
- Did they perform further laboratory tests, such as detection of serum level of BNP, NT-Pro-BNP, troponin, spiro(ergo)metry, imaging with ventilation-perfusion lung scan or HR-CT ?
- The authors did not mention the treatment of the rats – I suppose that they were without any hematological (anticoagulant) treatment...
The article could be edited after minor revision according to comments to the authors.
Author Response
Response to Reviewer 1 Comments
We would like to thank the reviewer for the constructive comments on the manuscript.
Point 1: “Did the authors test also full blood count and routine coagulation parameters, D-dimers or fibrin-degradation products? I consider them very important.”
Response 1: In this study, we evaluated standard hematological parameters in all animals. The data on blood parameters are now included in the Supplementary file A Table S1. Routine coagulation parameters, D-dimer or fibrin-degradation products, unfortunately, were not measured, which has been listed as the limitation. Please see changes (p.12, lines 287–288).
Point 2: “Did they perform further laboratory tests, such as detection of serum level of BNP, NT-Pro-BNP, troponin, spiro(ergo)metry, imaging with ventilation-perfusion lung scan or HR-CT ?”
Response 2: Imaging with ventilation-perfusion lung scanning and HR-CT were not used due to the high complexity of performing these techniques in small laboratory animals and minimal added value to complete layer-by-layer lung histology at all levels. BNP and NT-Pro-BNP are important biomarkers for the assessment of heart failure. We will certainly measure their levels in the future studies.
Point 3: “The authors did not mention the treatment of the rats – I suppose that they were without any hematological (anticoagulant) treatment...”
Response 3: Yes, the animals have not been treated with anticoagulants. Similar group is planned for the next work, where the microspheres will contain fibrin as a slow-release agent. The text has been modified accordingly (p. 13, lines 329–330).
Reviewer 2 Report
Introduction - is lacking the information to another publication by the authors that would be natural to site and for clarifying what is the new addition by the current study since the studies might seem to be very similar; European Heart Journal, Volume 40, Issue Supplement_1, October 2019, ehz745.0075, https://doi.org/10.1093/eurheartj/ehz745.0075
Introduction is lacking information to alginate, and the properties (both chemical and biological; ie fibrotic, thromboinflammatory) that do exist. Authors should give background knowledge to the alginate material of interest to create this model. Which chemical and biological properties makes the alginate interesting to create this model ?
Materials and methods:
Animals: There is ethical concerns to the large number of animals (rats) used as well as the level of suffering. There is explained that 44% of the animals given alginate micropsheres died within the first hour after injection; ie 44 rats. It is refered to National health institute (where and which guidelines)- but not to the three R`s (reduction, replacement, refinement). The authors should describe why there was a need for so many animals, and why it was considered to be within the ethical standards with this high amount of animal death (that suggest considerable suffering). Further there is also lacking sufficient description about the animal holding -please detail (now its only refered to as "were kept under standardized conditions ".
Alginate: Information about alginate (proportion of guluronic/mannuronic acid content), concentration, production conditions is fully lacking and should be detailed. What about endotoxin amounts? How did you measure the final microspheres diameter? Please specify. 2% barium solution (in buffer, saline, water ?), washing ?. How and in what did you perform the final concentration used for injection?
Experimental - injection: It is described as follows; " they were injected 8 times at 4-day intervals with 9367±551 MS suspended in 1 mL 297
of saline into the tail vein to reproduce PE. "
This seems to be a huge amount of micropsheres in the surviving animals. Why is this necessary, and how can you adjustify this as ethical ? There is not given data on how many survivors after each injection, which will be important information in considering this as a further hypertension model.
Choice of parametres for assessing inflammation and fibrosis; The authors chose endothelin, FGF-1, and CRP, but the explanations for these choices are only partly explained and the connections should be more detailed. Also, in the discussion part is emphazising several inflammatory cytokines as relevant. Why did not the authors also measure some of these cytokines? The discussion to cytokines would be more relevant if the current study had included the cytokines considered to be most relevant.
Statistics: Is somewhat unclear explained, particular the last part and should be reformulated; "... Subsequent analysis of differences in pairs was performed using the multiple comparison method according to the Kruskal-Wallis test for indicators with a statistically significant difference according to multivariate analysis; P-values <0.05 were considered significant."
Results:
Figure legends in general: the number of rats for each condition and time-point is lacking. Please specify.
Fig. 3: Histology - please specify the number of rats and histological sections examined. Description of the histological findings could be more detailed (ie; cell-types the first two weeks versus later time points seems to differ and could be described). In line 119-121 is explained; "In parallel with the biodegradation of microspheres, the severity of inflammatory
changes in the vascular wall and perivascular space decreased". This sentence is problematic in two ways - firstly - have you any prove of biodegradation? If there is any prove - how did you measure this and where is the data ? Secondly, "there severity of inflammation decreased" is not in a scientific language or content, as explained below;
Inflammation is a transient process, that eventually will be replaced by a chronic fase (in case the trigger persists as in case of a biomaterial). This might results in the replacement of granulocytes/monocytes with long-persisting cells (macrophages, fibroblasts, myofibroblasts) and collagen structures. This could be in consistence with the current observations. There exists thorough review litteratur both to biomaterials and specific to alginate. This is also recommended for the discussion part.
Discussion;
Line 221/222: Alginate is wrongly given as "Sodium alginate is a linear polysaccharide consisting of beta-D-mannuronic and alpha-L-hyaluronic acid residues..." Alginate contains guluronic acid. The authors are recommended to dive into the alginate litteratur, and there exists several well written reviews that could be helpful. This could also provide some background to discuss the issue of alginate degradation. The authors seems to anticipate that the microspheres are degraded. But - will the microspheres degrade ? How will the degradation be possible ? As to this reviwers knowledge there has been found no body enzymes that can degrade alginate. So - how did you make your microspheres design to promote "degradation" ?
There is lacking a discussion on the animal model to the three R`s. Would the model be robustly revealing differences ? From the data provided the differences seems to be to limited to serve as a basis for more complex testing of potential medicine of interest.
Author Response
Response to Reviewer 2 Comments
We would like to thank the reviewer for the fair criticism of the manuscript. Although some points are correctly criticized, we believe that the findings of this study are relevant to the scope of the journal and will be of interest to its readership.
Point 1: “Introduction - is lacking the information to another publication by the authors that would be natural to site and for clarifying what is the new addition by the current study since the studies might seem to be very similar; European Heart Journal, Volume 40, Issue Supplement_1, October 2019, ehz745.0075, https://doi.org/10.1093/eurheartj/ehz745.0075”
Response 1: The abstract mentioned describes the results of out pilot study, which compared two approaches to modeling CTEPH in rats: the use of native blood clots and artificial particles based on sodium alginate. In this separate study, limited number of observation time points and experimental end-points have been used. We have added this abstract to the reference list, and it is now mentioned in the Introduction (p. 2, line 59).
Point 2: “Introduction is lacking information to alginate, and the properties (both chemical and biological; ie fibrotic, thromboinflammatory) that do exist. Authors should give background knowledge to the alginate material of interest to create this model. Which chemical and biological properties makes the alginate interesting to create this model ?”
Response 2: Alginates are linear biopolymers consisting of 1,4-linked β-D-mannuronic acid (M) and 1,4 ɑ-L-guluronic acid (G) residues arranged in homogenous (poly-G, polyM) or heterogenous (MG) block-like patterns. Alginates can be easily formed into diverse semisolid or solid structures under mild conditions because of their unique ability of sol/gel transition. They may undergo in situ gelation that makes alginate materials promising tools for a wide range of applications, including injectable vehicles for tissue engineering or topical drug delivery systems. Among various alginates, sodium alginate is one of the most widely investigated ones in the pharmaceutical and biomedical field [Szekalska, M.; Puciłowska, A.; Szymańska, E.; Ciosek, P.; Winnicka, K. Alginate: Current Use and Future Perspectives in Pharmaceutical and Biomedical Applications. Int. J. Polym. Sci. 2016, 2016, 7697031]. Also, sodium alginate is regarded as biocompatible, nonimmunogenic, nontoxic and not degradable in mammalian gastro-intestinal tract material [Sachan, K. N.; Pushkar, S.; Jha, A.; Bhattcharya, A. Sodium alginate: the wonder polymer for controlled drug delivery. Journal of Pharmacy Research. 2009, 2, 1191–1199]. The text has been modified accordingly (p. 2, lines 63–72).
Point 3: “Animals: There is ethical concerns to the large number of animals (rats) used as well as the level of suffering. There is explained that 44% of the animals given alginate micropsheres died within the first hour after injection; ie 44 rats. It is refered to National health institute (where and which guidelines)- but not to the three R`s (reduction, replacement, refinement). The authors should describe why there was a need for so many animals, and why it was considered to be within the ethical standards with this high amount of animal death (that suggest considerable suffering). Further there is also lacking sufficient description about the animal holding -please detail (now its only refered to as "were kept under standardized conditions.”
Response 3: We used right ventricle systolic pressure (mmHg) on 6 weeks after the last injection of MSs as primary end-point of the study. The sample size per group was determined using the following parameters: SD value (± 7) on the basis of our pilot study, desired confidence level (95%), statistical power (0.9), effect size (13) calculated according to J. Cohen formula (Controls vs. CTEPH) (Power & Sample Size Program, version 3.1.6). According to the results of sample size calculation, n = 7 per time point has been found to be sufficient for making conclusions. The text has been modified accordingly (p. 16, lines 422–427).
Important details have been added to the description of care of laboratory animals: «The animals were maintained in individually ventilated cages on a 12 h light/dark cycle and were provided with food and water ad libitum. The temperature and humidity within the cages were kept within the ranges recommended by the Guide for the Care and Use of Laboratory Animals» (p. 13, lines 308–311).
Point 4: “Alginate: Information about alginate (proportion of guluronic/mannuronic acid content), concentration, production conditions is fully lacking and should be detailed. What about endotoxin amounts? How did you measure the final microspheres diameter? Please specify. 2% barium solution (in buffer, saline, water ?), washing ?. How and in what did you perform the final concentration used for injection? Experimental - injection: It is described as follows; " they were injected 8 times at 4-day intervals with 9367±551 MS suspended in 1 mL 297 of saline into the tail vein to reproduce PE.”
Response 4: In this study, we used "Alginic acid sodium salt from brown algae, BioReagent, suitable for plant cell culture, low viscosity" (cat. # A0682) (Sigma-Aldrich, USA). The reagent specification is attached to the letter. Microspheres were stabilized in a barium chloride solution, immediately before administration, they were washed in saline. The size and number of microspheres after encapsulation were determined in vitro using microscopy. To determine the number of microspheres, the initial dose was diluted 10 times, and the number of microspheres per unit volume was determined. The procedure was repeated three times to improve the measurement accuracy.
Point 5: “This seems to be a huge amount of micropsheres in the surviving animals. Why is this necessary, and how can you adjustify this as ethical ? There is not given data on how many survivors after each injection, which will be important information in considering this as a further hypertension model.”
Response 5: The table showing mortality rates during the modeling of CTEPH is now provided in the Supplementary file A Table S2.
In our pilot study (Karpov, A.; Anikin, N.; Cherepanov, D.; Mihailova, A.; Krasnova, M.; Smirnov, S.; et al. A new rat model of chronic thromboembolic pulmonary hypertension induced by repeated intravenous administration of biodegradable alginate microspheres. Eur. Heart J. 2019, 40, 1950), we used a protocol with a lower dose of microspheres for fewer injections (4 injections), which significantly increased the survival rate of animals, but has not led to sustained CTEPH. It should be noted that rats are quite resistant to development of CTEPH, which is confirmed by numerous unsuccessful attempts to model this pathology using native blood clots. In addition, sodium alginate microspheres are partially biodegradable, which mimics delayed thrombolysis in a person with CTEPH.
Point 6: “Choice of parametres for assessing inflammation and fibrosis; The authors chose endothelin, FGF-1, and CRP, but the explanations for these choices are only partly explained and the connections should be more detailed. Also, in the discussion part is emphazising several inflammatory cytokines as relevant. Why did not the authors also measure some of these cytokines? The discussion to cytokines would be more relevant if the current study had included the cytokines considered to be most relevant.”
Response 6: Undoubtedly, the small range of analyzed cytokines is a limitation of this work, which we have outlined in the limitation section. The emphasis on endothelin, CRP and FGF-1 was made due to the particular importance of endothelial dysfunction, aseptic inflammation, and proliferation of endothelial cells/fibroblasts, which are consistent with the functions of these cytokines (p. 12, line 288).
Point 7: “Statistics: Is somewhat unclear explained, particular the last part and should be reformulated; "... Subsequent analysis of differences in pairs was performed using the multiple comparison method according to the Kruskal-Wallis test for indicators with a statistically significant difference according to multivariate analysis; P-values <0.05 were considered significant."”
Response 7: Description of statistical methods has been modified for clarity: «The Kruskal‐Wallis test was performed to determine differences in outcomes, followed by pairwise inter‐group comparisons by non‐parametric Mann‐Whitney U test; P values ≤ 0.05 were considered to indicate statistical significance. All data are expressed as mean ± standard deviation.» (p. 16, lines 427–430).
Point 8: “Figure legends in general: the number of rats for each condition and time-point is lacking. Please specify.”
Response 8: The number of animals has been specified in the figure legends.
Point 9: “Fig. 3: Histology - please specify the number of rats and histological sections examined. Description of the histological findings could be more detailed (ie; cell-types the first two weeks versus later time points seems to differ and could be described). In line 119-121 is explained; "In parallel with the biodegradation of microspheres, the severity of inflammatory changes in the vascular wall and perivascular space decreased". This sentence is problematic in two ways - firstly - have you any prove of biodegradation? If there is any prove - how did you measure this and where is the data ? Secondly, "there severity of inflammation decreased" is not in a scientific language or content, as explained below.”
Response 9: The number of rats at each time point was 7. We analyzed two cross-sections from each rat. We analyzed all the vessels on the section related to the branches of the pulmonary artery. Explanations on the number of animals at each time point are included in the text of the manuscript.
To assess the rate of biodegradation of microspheres in vivo at each point of excretion, the percentage of detected microspheres on two distal sections of the lung was calculated. The number of microspheres detected at week 0 immediately after the administration of the last dose of microspheres was taken as 100%. The scheme of assessment of MS biodegradation in vivo are now included in the Supplementary file B Figure S1.
The sentence "In parallel with the biodegradation of microspheres, the severity of inflammatory changes in the vascular wall and perivascular space decreased" has been deleted from the manuscript as it lacks sufficient scientific evidence.
Point 10: “Inflammation is a transient process, that eventually will be replaced by a chronic fase (in case the trigger persists as in case of a biomaterial). This might results in the replacement of granulocytes/monocytes with long-persisting cells (macrophages, fibroblasts, myofibroblasts) and collagen structures. This could be in consistence with the current observations. There exists thorough review litteratur both to biomaterials and specific to alginate. This is also recommended for the discussion part.”
Response 10: Indeed, intravascular alginate microspheres to some extent can stimulate the activation of macrophages, fibroblasts and myofibroblasts [Veiseh, O.; Doloff, J.C.; Ma, M.; Vegas, A.J.; Tam, H.H.; Bader, A.R.; et al. Size- and shape-dependent foreign body immune response to materials implanted in rodents and non-human primates. Nat. Mater. 2015, 14, 643-651; Vegas, A.J.; Veiseh, O.; Doloff, J.C.; Ma, M.; Tam, H.H.; Bratlie, K.; et al. Combinatorial hydrogel library enables identification of materials that mitigate the foreign body response in primates. Nat. Biotechnol. 2016, 34, 345-352]. The same cellular response is, however, typical in the situation of persistent thrombus in CTEPH. It should be noted that the number of preserved microspheres by 18 weeks was 2.4% of those available at 0 time point. Thus, fibrotic changes in the vascular wall cannot be explained solely by the response of the immune system and fibroblasts to the biomaterial. Consideration should be given to the role of persistent increases in pulmonary artery pressure and endothelial dysfunction.
Point 11: “Line 221/222: Alginate is wrongly given as "Sodium alginate is a linear polysaccharide consisting of beta-D-mannuronic and alpha-L-hyaluronic acid residues..." Alginate contains guluronic acid. The authors are recommended to dive into the alginate litteratur, and there exists several well written reviews that could be helpful. This could also provide some background to discuss the issue of alginate degradation. The authors seems to anticipate that the microspheres are degraded. But - will the microspheres degrade ? How will the degradation be possible ? As to this reviwers knowledge there has been found no body enzymes that can degrade alginate. So - how did you make your microspheres design to promote "degradation" ?”
Response 11: According to the literature, the question of biodegradation of alginate is controversial. On the one hand, there is evidence of good biodegradation [Pawar, S.N.; Edgar, K.J. Chemical modification of alginates in organic solvent systems. Biomacromolecules. 2011, 12, 4095-4103]; on the other hand, it is indicated that there are no enzymes that break down alginate polymer chains in mammals [Lee, K.Y.; Mooney, D.J. Alginate: properties and biomedical applications. Prog. Polym. Sci. 2012, 37, 106-126]. Ionically cross-linked alginate gels can be dissolved by release of the divalent ions cross-linking the gel into the surrounding media due to exchange reactions with monovalent cations such as sodium ions [Lee, K.Y.; Mooney, D.J. Alginate: properties and biomedical applications. Prog. Polym. Sci. 2012, 37, 106-126]. Our data are more likely in favor of appropriate biodegradation and elimination of alginate, since we see neither its accumulation in macrophages nor granulomas. The text has been modified accordingly (p. 11, lines 240–246).
Point 12: “There is lacking a discussion on the animal model to the three R`s. Would the model be robustly revealing differences ? From the data provided the differences seems to be to limited to serve as a basis for more complex testing of potential medicine of interest.”
Response 12: This model, unfortunately, is characterized by a significant death of animals after the administration of MS, which is not entirely compatible with the requirements of 3R. Nevertheless, we consider it ethically acceptable to use this model, since it is impossible to achieve the desired result in any other way, and patients with CTEPH need new effective approaches to treatment. The text has been modified accordingly (pp 12-13. lines 291–294).
Using the described model, we tested JAK inhibitor for the prevention and treatment of CTEPH (unpublished data). The use of test compound has led to a significant decrease in the level of the hypertrophy index of vascular wall and an increase in exercise tolerance. Therefore, we believe that the differences in the key end-points between controls and CTEPH animals are significant enough and leave room for improvement with therapeutic agents.
Round 2
Reviewer 2 Report
The comments and changes are adequate for acceptance of the manuscript.
Some comments to changes that still would be warranted;
The alginate is now described (which is fine) - but references to the first part is totally lacking. These statements needs to have references, either original or to proper reviews.
In M&M section the description of alginate is still scarce. No description of M/G content is present, which is essential for the biodegradation that the authors anticipate is the reason for reduction over time. Biodegradation would also be dependent on the type of gelling ions, which would therefor also be important moment for discussion. In this respect, one can also wonder why the authors are not using calcium as gelling ions, if microspheres dissolving is a warranted effect.
The authors states that they could not see alginate taken up by macrophages - but did they have relevant methods for detecting this?
The inclusion of the ethical concerns are important to a study that undoubtedly causes animal suffering. An description of the animal monitoring included behavior would further strengthen this part.
To increase the impact/scientific value of the manuscript there is a need more details to the alginate properties both of relevance to biodegradation as well as blood compatibility aspects. The manuscript as it is lacking relevant and important literature of the field.
Author Response
Response to Reviewer 1 Comments
We would like to thank the reviewer for the constructive comments on the manuscript.
Point 1: “The alginate is now described (which is fine) - but references to the first part is totally lacking. These statements needs to have references, either original or to proper reviews.”
Response 1: According to the Reviewer’s advice, we have changed the description of alginates and provided new references. The modified part of the text (p.2, lines 63–70) is provided below:
“Alginates are linear biological polymers composed of 1,4-linked β-D-mannuronic acid (M) and 1,4 ɑ-L-guluronic acid (G) residues arranged in homogenous (poly-G, poly-M) or heterogenous (MG) block-like patterns [Rastogi, P., Kandasubramanian, B. Review of alginate-based hydrogel bioprinting for application in tissue engineering. Biofabrication. 2019, 11, 042001; Ueno, M., Oda, T. Biological activities of alginate. Adv. Food Nutr. Res. 2014, 72, 95-112; Ching, S.H., Bansal, N., Bhandari, B. Alginate gel particles-A review of production techniques and physical properties. Crit. Rev. Food Sci. Nutr. 2017, 57, 1133-1152]. Owing to their unique capability of sol-gel phase transition, alginates can easily form various semisolid or solid structures. Further, alginates undergo in situ gelation that makes them promising materials for a wide range of applications, including injectable biopolymers for tissue engineering or targeted drug delivery systems [Axpe, E., Oyen, M.L. Applications of Alginate-Based Bioinks in 3D Bioprinting. Int. J. Mol. Sci. 2016, 17, 1976. Hariyadi, D.M., Islam, N. Current Status of Alginate in Drug Delivery. Adv. Pharmacol. Pharm. Sci. 2020, 2020, 8886095]. Sodium alginate is one of the most extensively studied types of alginate both in the pharmaceutical and biomedical field [17].”
Point 2: “In M&M section, the description of alginate is still scarce. No description of M/G content is present, which is essential for the biodegradation that the authors anticipate is the reason for reduction over time. Biodegradation would also be dependent on the type of gelling ions, which would therefor also be important moment for discussion. In this respect, one can also wonder why the authors are not using calcium as gelling ions, if microspheres dissolving is a warranted effect.”
Response 2: The M/G ratio is 1.56. Please see changes (p.13, line 325).
In pilot studies, we tested calcium as gelling ions alone or in combination with barium ions (1: 3, 1: 1, 3: 1). However, the obtained microspheres did not have sufficient stability in the circulation and vascular lumen.
Point 3: “The authors states that they could not see alginate taken up by macrophages - but did they have relevant methods for detecting this?”
Response 3: Unfortunately, we have not used specific methods to assess the accumulation of alginate in macrophages. However, by 6 and 18 weeks after the administration of microspheres, we have not observed the accumulation of macrophages in the perivascular region and the appearance of multinucleated giant cells, which is usually observed when the removal of foreign material is impaired.
Point 4: “The inclusion of the ethical concerns are important to a study that undoubtedly causes animal suffering. An description of the animal monitoring included behavior would further strengthen this part.”
Response 4: The causes of death of animals in all cases were ischemic stroke and acute heart failure. The rats died within 2-3 minutes from the moment of microsphere injection. The death of animals in the period after the last injection of microspheres was not observed. The animals' condition was monitored by an experienced veterinarian by clinical examination once a day. However, the surviving animals did not show any abnormalities.
Please see changes (p.2, lines 79-81).
Point 5: “To increase the impact/scientific value of the manuscript there is a need more details to the alginate properties both of relevance to biodegradation as well as blood compatibility aspects. The manuscript as it is lacking relevant and important literature of the field.”
Response 5: We agree with the Reviewer and we have modified the text accordingly. Please see changes (p.11, lines 232–242).
